# Preoperative Axillary Ultrasound in the Era of Z0011: A Model for Predicting High Axillary Disease Burden

**DOI:** 10.3390/curroncol32060307

**Published:** 2025-05-27

**Authors:** Ashley DiPasquale, Lashan Peiris

**Affiliations:** 1Department of Surgery, University of Alberta, Edmonton, AB T6G 2B7, Canada; 2Comprehensive Breast Center of Arizona, Phoenix, AZ 85016, USA

**Keywords:** breast cancer, axillary management, ultrasound

## Abstract

The ACOSOG Z0011 and IBCSG 23-01 trials demonstrated that axillary lymph node dissection (ALND) offers no prognostic benefit in breast cancer patients with clinically negative axillae and low disease burden (one to two positive nodes) on sentinel lymph node biopsy (SLNB). However, uncertainty remains regarding the management of patients with clinically negative axillae (cN0) who are found to have suspicious lymph nodes on imaging that are subsequently confirmed positive by biopsy. The current practice often directs these patients to upfront ALND, potentially exposing them to unnecessary surgical morbidity. This study aimed to assess the role of axillary ultrasound in predicting high axillary nodal burden and guiding surgical management. Using the Alberta Cancer Registry, we identified 107 cN0 breast cancer patients from 2010 to 2017 who underwent preoperative axillary ultrasound with positive biopsy followed by ALND. Our findings reveal that 42% of these patients had low axillary nodal burden on final pathology, meeting Z0011 criteria, and might potentially have avoided ALND. Furthermore, axillary ultrasound findings were not predictive of high axillary burden. These results highlight that many patients undergoing upfront ALND based on positive ultrasound-guided biopsy could benefit from SLNB alone. This supports the 2023 NCCN guidelines advocating for more selective use of ALND to minimize overtreatment and associated morbidity.

## 1. Introduction

Breast cancer is among the top three most common malignancies globally. One in eight North American women will develop invasive breast cancer in their lifetime [1]. The management of breast cancer has evolved greatly over the past two decades with ‘de-escalation’ of breast and axillary surgery and now strives to balance the avoidance of overtreatment with lowering rates of locoregional failure.

Historically, axillary lymph node dissection (ALND) at levels I and II was the standard of care for patients with invasive breast cancer. The advent of the less invasive sentinel lymph node biopsy (SLNB) has meant that patients can avoid the morbidity of an ALND [2]. ACOSOG Z0011 and IBCSG 23-01 studies demonstrated that ALND has no prognostic benefit in patients who have clinically node-negative breast cancer (cN0) and have one to two positive metastatic nodes on sentinel node biopsy. Importantly, axillary ultrasound was not a mandatory part of the ACOSOG Z011 trial protocol. Patients were only excluded if they had “palpable adenopathy” on physical examination [3,4]. In response to these studies, ALND has been largely replaced by sentinel lymph node biopsy in cN0 patients.

Axillary ultrasound is now widely performed as part of the standard workup of early breast cancer in cN0 patients. As a result, the wording of various guidelines and tumor board discussions has caused some confusion with regards to what is classified as a “clinically node-negative axilla”. Is image-detected axillary disease, not apparent on physical exam, classified as ‘clinically node positive’? Alternatively, is a clinically negative axilla based solely on there being no palpable disease on physical exam—irrespective of what axillary imaging shows? Patients who are found to be node negative by physical examination but positive by radiology often undergo lymph node FNA or core biopsy. At the time of this study, in our center, if the lymph node cytology/biopsy is positive for malignancy, then many of these patients will undergo ALND, without having an SLNB (Figure 1).

Previous studies have shown that up to 40% of patients with a low tumor burden on preoperative workup have zero to two metastatic nodes after ALND. These patients would have met Z0011 criteria and been appropriate for SLNB only, avoiding the increased morbidity associated with ALND [5,6,7]. Lymphedema is seen in 3–13% of patients, and over 50% of patients continue to experience paresthesia 3 years after ALND [6,7,8]. Approximately 30% of patients experience chronic pain and restricted range of arm motion after the procedure [6].

Therefore, it is important to determine preoperatively which patients who are node negative on physical examination but node positive by imaging criteria are likely to have a low axillary burden to avoid an unnecessary ALND and the accompanying morbidity. Our study aimed to create and validate a predictive model that obtains estimates of a patient’s risk of high axillary nodal burden based on their axillary ultrasound findings and disease characteristics. Our goal was to help determine the usefulness of axillary ultrasound in cN0 patients and help surgeons better ‘triage’ which cN0 patients with a positive preoperative axillary lymph node biopsy might still be appropriate for an SLNB followed by selective completion ALND (cALND) as appropriate.

## 2. Materials and Methods

### 2.1. Study Design and Population

This retrospective cohort study evaluated axillary nodal burden in women with early-stage breast cancer and preoperative ultrasound-guided biopsy-proven axillary nodal metastases who subsequently underwent axillary lymph node dissection (ALND). Eligible patients were identified from the Alberta Cancer Registry, a comprehensive population-based database that captures all cancer diagnoses in the province of Alberta, Canada. Inclusion criteria resulted in the inclusion of female patients aged 18 years or older, diagnosed between 1 January 2007 and 31 December 2017, with clinically node-negative (cN0) invasive breast cancer (T1–T3) on physical examination. All included patients had a suspicious lymph node identified on preoperative axillary ultrasound and a positive needle biopsy (fine-needle aspiration or core biopsy) followed by definitive surgery—either mastectomy or breast-conserving surgery—with ALND.

Patients were excluded if they had any of the following: stage IV disease at presentation, inflammatory breast cancer, recurrent breast cancer, a history of prior ipsilateral axillary surgery, receipt of neoadjuvant chemotherapy, or treatment before publication of the ACOSOG Z0011 trial in 2010. The study was approved by the Covenant Health Research Ethics Board, and all procedures were performed in accordance with institutional and international ethical guidelines, including the Declaration of Helsinki (1975, as revised in 2009).

### 2.2. Data Collection and Variables

Data were collected from both the Alberta Cancer Registry and a retrospective review of electronic medical records, including operative reports, pathology reports, and imaging records. Supplementary chart review was performed to ensure completeness of data for variables not captured within the registry.

Collected variables included demographic and clinical data (age at diagnosis, body mass index [BMI], and surgical procedure), as well as tumor characteristics (clinical tumor size, multifocality, histologic type, histologic grade, estrogen receptor [ER] status, progesterone receptor [PR] status, and human epidermal growth factor receptor 2 [HER2] status).

Preoperative axillary ultrasound variables included number of suspicious lymph nodes visualized (<3 vs. ≥3), longest diameter of the most suspicious lymph node (<1 cm vs. ≥1 cm), cortical thickness (<3 mm vs. ≥3 mm), and replacement of nodal hilum (>50% vs. preserved). If the cortex or hilum was not reported as abnormal, it was considered normal. All ultrasound studies were interpreted by fellowship-trained breast radiologists. To maintain confidentiality, facility identifiers were anonymized.

Patients were categorized into 2 groups based on final surgical pathology after ALND: low axillary burden: 0–2 positive lymph nodes; or high axillary burden: ≥3 positive lymph nodes.

### 2.3. Statistical Analysis

Descriptive statistics were used to summarize baseline characteristics. Continuous variables are presented as means with standard deviations and categorical variables as frequencies and percentages. Univariate analysis was performed to assess the association between each variable and axillary nodal burden using either the χ^2^ test or Fisher’s exact test, as appropriate.

A multivariable logistic regression model was developed using forward stepwise selection. Variables were selected a priori based on clinical significance and the prior literature. Odds ratios (ORs) and 95% confidence intervals (CIs) are reported. The discriminatory power of the model was evaluated using a receiver operating characteristic (ROC) curve, and predictive accuracy was quantified using the area under the curve (AUC). Statistical significance was set at a 2-tailed *p*-value < 0.05. All analyses were conducted using Stata/IC, version 15.1 (StataCorp, College Station, TX, USA).

## 3. Results

### 3.1. Patient Characteristics

The Alberta Cancer Registry identified 2947 patients who had ALND performed between 2007 and 2017. After eliminating patients with recurrent breast cancer and those who received neoadjuvant therapy, 813 patients remained. One hundred and seven patients met our final inclusion criteria.

Of these, 45 patients had one to two positive nodes on final pathology and 62 patients had three or more positive nodes on final pathology after ALND (Figure 2). This shows that, in our local data, 42% of cN0 patients with a positive lymph node on US-guided biopsy had low axillary nodal burden. These results remained consistent after analyzing the 73 patients who had T1–T2 tumors. Of these patients, 31 patients, 43%, would have met Z0011 criteria and could have avoided an ALND.

Patient characteristics are listed in Table 1. There were no statistically significant differences in baseline characteristics between the two groups including size, grade, presence of LVI, hormone receptor status, Her2 status, axillary ultrasound findings, and operating facility.

### 3.2. Predictive Model

A multiple logistic regression model was calculated to predict high axillary nodal burden based on a longest node diameter greater than 1 cm, lymph node cortex thickening > 3 mm, a replaced fatty hilum, tumor grade, LVI presence, hormone receptor status, HER2 status, and age > 50 years old. A significant regression equation was found with an N of 106, X2 of 6.90, and *p* = 0.008. The presence of LVI was the only parameter found to be predictive of high axillary nodal burden. Patients with LVI had 3.2 increased odds of having high axillary nodal burden with a *p*-value of 0.010 and 95% CI of 1.3–7.8. The area under the ROC was 0.616 (Figure 3). The model had a sensitivity of 82.3% and a specificity of 40.9%. Because of the poor performance of the model, validation of the model was abandoned.

## 4. Discussion

The goal of our study was to help determine the usefulness of axillary ultrasound in cN0 patients and to help surgeons better triage which patients who are deemed negative on clinical exam but shown to be radiographically node positive might still be appropriate for an SLNB followed by completion ALND as appropriate.

Our findings revealed that 42% of the cohort had a low axillary nodal burden on final pathology. Specifically, among cN0 patients with T1–T2 tumors, 45% had only one to two positive lymph nodes and may have been suitable candidates to avoid ALND. Furthermore, our predictive model demonstrated that axillary ultrasound was not a reliable tool for estimating the risk of high axillary nodal burden. The literature shows that this high rate of low axillary nodal burden in patients triaged directly to ALND is not isolated to our local data. Morrow et al. found that 47% of patients with a positive LN diagnosed by US and needle biopsy before surgery had only one to two total positive LNs on final pathology. They concluded that 18.4% could have been safely managed with SLNB alone if treated according to ACOSOG Z0011 criteria [9]. Boland et al. found that 38.6% of patients with a positive lymph node biopsy had fewer than three positive nodes, and 7.8% met Z0011 eligibility criteria, suggesting they might have avoided ALND [10]. Similarly, Caudle et al. reported that 52% of cN0 patients with a positive lymph node needle biopsy had fewer than three positive nodes on final ALND pathology [11]. Likewise, data from the Mayo Clinic showed that 48.4% of patients with axillary metastases identified on ultrasound had two or fewer positive nodes on surgical pathology [12].

Previous studies that attempted to predict high axillary nodal burden found that the number of abnormal lymph nodes found on ultrasound was the most predictive of high axillary burden. Lim et al. found that more than three abnormal lymph nodes found on sonography predicted high axillary nodal burden with an odds ratio of 20.7 [13]. Liu et al. found a significantly greater proportion of women with high nodal burden had more than one abnormal LN identified on their preoperative axillary US [14]. In another study, all patients with more than three metastatic LNs had more than three suspicious LNs on US imaging [15]. The recent NCCN 2023 recommendations for axillary staging stating that patients with no palpable lymph node at diagnosis, fewer than two suspicious lymph nodes on imaging, or fewer than two positive lymph nodes confirmed by needle biopsy ± clip placement can undergo SLNB followed by cALND when appropriate based on Z0011 and IBCSG 23-01 [16]. This also aligns with the ASCO and other regional guidelines [17,18].

Further supporting the ongoing de-escalation of axillary surgery, the SENOMAC trial demonstrated that completion axillary lymph node dissection (ALND) can be safely omitted in patients with clinically node-negative (cN0) breast cancer who have one or two sentinel node macrometastases [19]. The trial required preoperative axillary ultrasound and allowed inclusion of cN0 patients with biopsy-proven nodal metastasis of suspicious nodes found on ultrasound. The trial also allowed inclusion of patients with additional micrometastases and extracapsular extension. Although the SENOMAC trial did not quantify total nodal burden beyond the sentinel nodes, its findings reinforce that even patients with biopsy-proven nodal disease preoperatively may be safely managed with SLNB alone, provided the final pathology confirms low axillary burden. This aligns with our study’s findings and supports a growing body of evidence advocating for tailored, less morbid surgical approaches in appropriately selected patients. However, it is noted that many patients in the SENOMAC study did receive nodal radiation.

The ongoing TAXIS trial (Tailored Axillary Surgery with or without Axillary Lymph Node Dissection in Patients with Node-Positive Breast Cancer) is further exploring the potential to safely de-escalate axillary surgery in select patients with node-positive disease [20]. This multicenter, randomized phase III study is evaluating whether tailored axillary surgery (TAS) can be combined with axillary radiotherapy to avoid the morbidity of completion ALND without compromising oncologic outcomes. This trial is still ongoing but is expected to provide level I evidence on whether ALND is necessary in this patient population.

In addition to evolving surgical and radiation strategies, systemic therapies are also reshaping axillary management in early breast cancer. The emergence of CDK4/6 inhibitors such as ribociclib, particularly in hormone-receptor-positive, HER2-negative disease, offers potential benefits in patients with nodal involvement. The recent NATALEE trial demonstrated improved disease-free survival with ribociclib in patients with stage II or III breast cancer, suggesting that enhanced systemic control may further justify de-escalation of axillary surgery in appropriately selected cases [21]. Although our study did not assess systemic therapy usage, these advances underscore the importance of integrating multidisciplinary treatment planning.

While axillary ultrasound (AUS) may not be a definitive predictor of a high axillary burden, it remains a valuable tool for identifying patients who align with current NCCN axillary staging criteria [22,23]. Moreover, looking ahead, AUS could assume even greater importance, particularly in light of the recent findings from the SOUND and INSEMA trials [24,25]. The study’s authors concluded that patients with small breast cancer and sonographically normal-appearing lymph nodes can safely forgo any axillary surgery, provided that the absence of pathological information does not impact the postoperative treatment plan [26].

Other factors associated with an increased risk of high nodal disease burden include high tumor grade, LVI, and Her2 positivity [27,28]. A smaller primary tumor, invasive lobular carcinomas, and low Ki67 expression were associated with a decreased risk of high nodal burden [27]. In contrast, additional studies, including ours, have shown that breast tumor histology, grade, ER or PR positivity, and Her2 status do not statistically significantly increase the risk of high nodal disease burden [13].

While our findings are consistent with and reinforce the existing literature, our study offers several unique contributions that enhance its relevance to modern clinical practice. For example, Pilewskie et al. demonstrated that, among clinically node-negative patients with abnormal axillary imaging, 71% ultimately did not meet criteria for ALND and were spared the associated surgical morbidity. Their study concluded that abnormal axillary findings on ultrasound, MRI, or mammography were not reliable indicators for ALND; however, all patients in that cohort underwent sentinel lymph node biopsy (SLNB) and were subsequently triaged to completion ALND only if warranted [29]. Importantly, that study did not analyze specific ultrasound characteristics in detail. In contrast, prior investigations by Caudle and Boland suggested a high likelihood of ALND in biopsy-positive patients, reinforcing the notion that preoperative nodal involvement generally necessitates extensive axillary surgery [10,11].

Our study specifically examines a distinct population of cN0 patients with biopsy-proven nodal metastases identified via ultrasound-guided needle biopsy—a group traditionally excluded from trials such as Z0011 and commonly triaged directly to ALND. Notably, we found that nearly half of these patients had a low nodal burden (defined as fewer than two positive nodes) on final pathology, suggesting that many may have been overtreated with full axillary dissection. Furthermore, Morrow et al. did not routinely incorporate axillary imaging and limited ALND to predefined SLNB criteria, while our analysis incorporated detailed preoperative ultrasound features—such as cortical thickness and the number of suspicious nodes—along with patient and tumor characteristics [9]. We introduced a predictive model using these preoperative sonographic features, which we hoped would serve as a practical tool for identifying patients at higher risk of extensive nodal involvement and potentially guide selective use of ALND in this previously undertreated or overtreated subgroup.

Our model was intentionally built using only variables available prior to surgery—including patient demographics, tumor characteristics, and detailed preoperative ultrasound findings—all of which are routinely collected in standard clinical practice. This approach aimed to provide a practical, non-invasive method to predict high axillary burden. However, our results demonstrated that preoperative variables alone were not sufficiently predictive of high nodal burden.

The limitations of our study included the inability to control for inter-observer variability between radiologists and pathologists. To help mitigate this, axillary ultrasound (AUS) was performed across multiple centers, with interpretations by various breast radiologists, and all axillary lymph node dissections (ALNDs) were performed at sites throughout Alberta, Canada. Another limitation was our small sample size, which did not allow us to control for all potential confounding variables in the model. However, univariate analysis demonstrated no statistically significant differences between the two groups in any variable except for lymphovascular invasion (LVI), which was accounted for in our multivariate model. The small sample size was not unexpected, as, in the post-Z011 era, most patients with early-stage breast cancer undergo sentinel lymph node biopsy (SLNB) rather than ALND. Moreover, many patients with node-positive disease are now referred for neoadjuvant chemotherapy and would thus have been excluded from our study. Patients with clinically node-positive (cN1) disease were also excluded, as SLNB is not indicated in the setting of palpable axillary disease. Additionally, this study did not assess imaging data from MRI or PET/CT scans, which may have value in predicting a high axillary disease burden when used in combination with AUS.

Axillary lymph node dissections are not without morbidity. The complications of the procedure include lymphedema, restrictive arm ROM, chronic pain, and paresthesia, and all at much higher rates than seen after SLNB [6,7,8,28]. Our study adds to the growing body of literature suggesting that cN0 patients triaged directly to ALND because of axillary ultrasound findings may be overtreated. These results further support the NCCN 2023 recommendations for axillary staging stating that patients with no palpable lymph node at diagnosis, fewer than two suspicious lymph nodes on imaging, or fewer than two positive lymph nodes confirmed by needle biopsy ± clip placement can undergo SLNB followed by cALND when appropriate based on Z0011 and IBCSG 23-03 [16]. The ASCO guidelines support this approach, recommending that patients who are clinically node negative on physical examination but have sonographically abnormal lymph nodes on imaging—with or without confirmatory biopsy—be offered sentinel lymph node biopsy (SLNB) as the initial method of axillary staging. Although AUS may not be a reliable predictor of high axillary burden, the procedure is still useful to help determine which patients meet the current NCCN and ASCO axillary staging criteria as well as SOUND and INSEMA trial criteria.

## 5. Conclusions

In summary, our study highlights the limitations of axillary ultrasound (AUS) in accurately predicting high axillary nodal burden in cN0 breast cancer patients. Despite radiographic evidence of nodal involvement, a significant proportion of patients were found to have low axillary disease burden on final pathology and may have been appropriate candidates for SLNB alone. These findings are consistent with the existing literature and support the updated NCCN 2023 and ASCO guidelines, which advocate for SLNB in patients with fewer than two suspicious or biopsy-confirmed positive nodes. While AUS may not reliably estimate nodal burden, it remains an important tool for identifying patients who meet current axillary staging criteria and may have an evolving role as management strategies shift, especially in light of emerging evidence such as that from the SOUND and INSEMA trials. By refining our use of AUS and integrating clinical, pathological, and imaging factors, we can better individualize axillary management and potentially spare patients the morbidity associated with unnecessary ALND.

## Figures and Tables

**Figure 1 curroncol-32-00307-f001:**
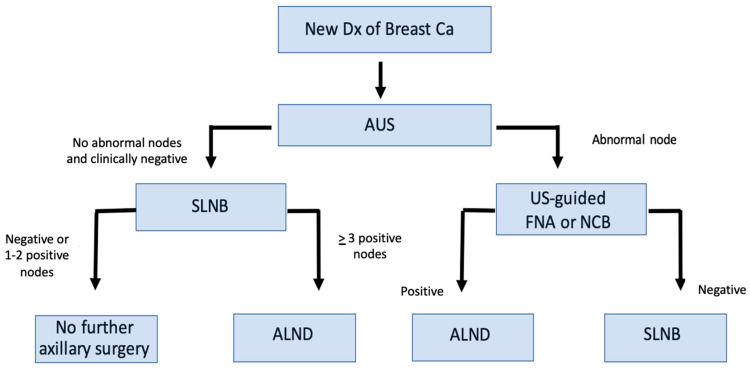
Flow chart showing axillary staging pathway for patients in our study after Z0011.

**Figure 2 curroncol-32-00307-f002:**
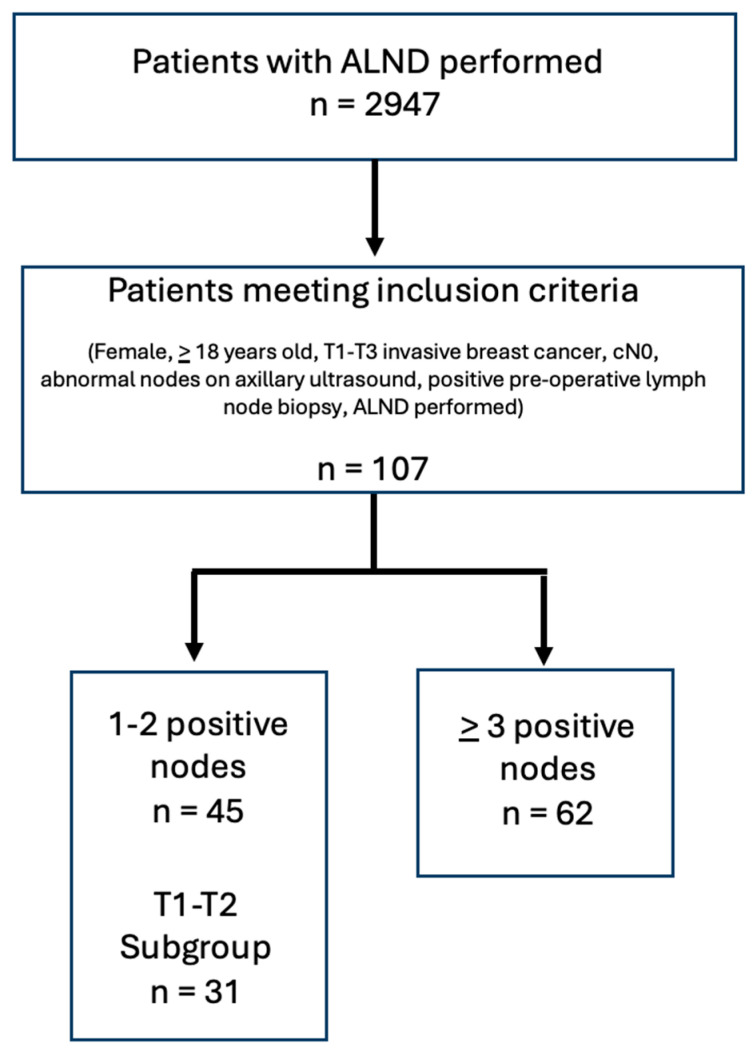
Patient selection and classification by axillary nodal burden.

**Figure 3 curroncol-32-00307-f003:**
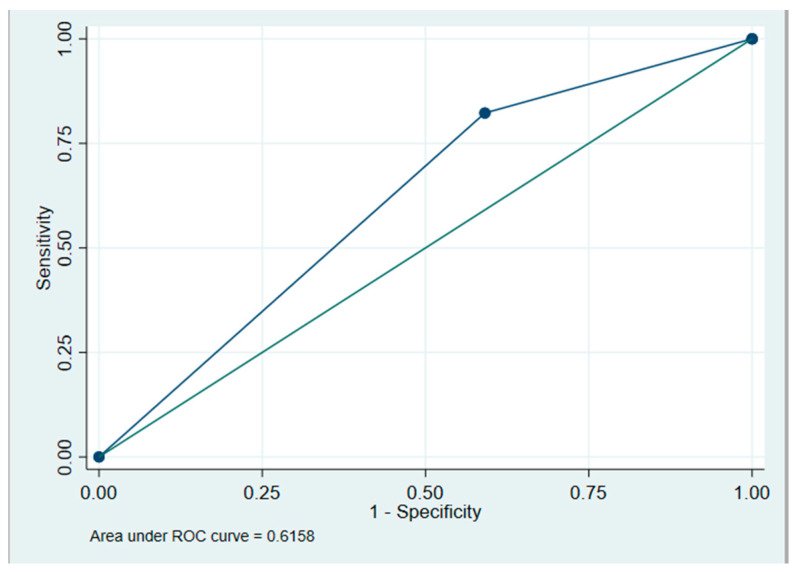
Area under the ROC.

**Table 1 curroncol-32-00307-t001:** Patient characteristics.

	Low Axillary Node Burden N = 45	High Axillary Node Burden N = 62	*p*-Value
Age at diagnosis [SD]	60.3 [10.9]	63.0 [15.0]	0.29
Age at diagnosis			0.88
	10 (22)	13 (21)	
	35 (78)	49 (79)	
BMI status			0.66
<18.5	2 (6)	1 (2)	
18.5–24.9	13 (29)	15 (24)	
25.0–29.9	11 (31)	17 (32)	
30.0–34.9	4 (11)	10 (19)	
≥35.0	8 (23)	8 (15)	
Facility *			0.36
1	19 (42)	19 (31)	
2	5 (11)	5 (8)	
3	3 (7)	8 (13)	
4	3 (7)	0	
5	0	1 (2)	
6	1 (2)	2 (3)	
7	1 (2)	6 (10)	
8	5 (11)	5 (8)	
9	4 (9)	4 (6)	
10	3 (7)	8 (13)	
11	1 (2)	1 (2)	
12	0	2 (3)	
13	0	1 (2)	
Invasive type			0.86
Ductal	40 (89)	56 (90)	
Lobular	4 (9)	4 (6)	
Mixed	1 (2)	2 (3)	
Tumor size (cm)			0.28
<2 cm	15 (38)	15 (25)	
2–5	16 (41)	27 (44)	
>5	8 (21)	19 (31)	
Focality			0.28
Unifocal	36 (88)	47 (77)	
Multifocal	4 (10)	8 (13)	
Multicentric	1 (2)	6 (10)	
Surgery performed			0.82
BCS	19 (44)	26 (42)	
Mastectomy	24 (56)	36 (56)	
Grade			0.69
1	5 (11)	8 (13)	
2	14 (32)	15 (24)	
3	25 (57)	39 (63)	
Hormone receptor status			0.05
Positive	0	6 (15)	
Negative	24 (100)	35 (85)	
Her2 status			0.67
Positive	16 (80)	30 (75)	
Negative	4 (20)	10 (25)	
Nodes on ultrasound			0.16
1	36 (80)	42 (68)	
≥2	9 (20)	20 (32)	
Longest diameter measurement (cm)	14.9 [8.7]	14.2 [7.0]	0.71
Longest diameter measurement (cm)			0.43
≤1 cm	14 (31)	15 (24)	
>1 cm	31 (69)	47 (76)	
Thickened cortex			0.12
<3 mm	18 (40)	16 (26)	
≥3 mm	27 (60)	46 (74)	
Replaced hilum **			0.18
Yes	37 (82)	44 (71)	
No	5 (18)	7 (29)	
Average number of nodes removed	14.8 [6.3]	15.7 [6.3]	0.47
LVI	26 (58)	51 (82)	0.005

BMI = body mass index; BCS = breast-conserving surgery; Her2 = human epidermal growth factor receptor 2; LVI = lymphovascular invasion. * Facility indicates surgical centers and includes inpatient hospitals and outpatient surgical centers. ** Fatty tissue normally found in the lymph node central hilum is absent or replaced by dense tissue.

## Data Availability

The raw data supporting the conclusions of this article will be made available by the authors on request.

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
