# Peer review of "Preoperative Axillary Ultrasound in the Era of Z0011: A Model for Predicting High Axillary Disease Burden"

_curroncol, 2025, doi:10.3390/curroncol32060307_

Round 1
Reviewer 1 Report
Comments and Suggestions for Authors
The authors present a retrospective multicenter analysis based on the Alberta Cancer Registry. The include 107 patients who were found to be clinically node-negative but node positive by ultrasound and fine-needle aspiration or core-neede biopsy. A predictor of 3 or more positive lymph nodes upon axillary dissection was lymphovascular invasion.
This is a confirmatory study adding to the existing literature. The authors use a large cancer registry, having the benefit of being representative of a real-world setting, yet the downside of having limited information available.
I was positively surprised to see that apparently the number of suspicious lymph nodes on ultrasonography (<3 nodes vs. >=3 nodes) were recordded in the registry as they are not recorded in many larger cancer centers.
If the data are available to the authors, I would find it interesting to know, how many patients received a breast MRI and/or PET-CT; again, of as much detailed information concerning those imaging modalities would be available as it seems to be the case for axillary ultrasound, it would be interesting to know, whether results of these diagnostic tools would predict higher nodal burden (however, this could be a novel study and therefore be out of scope for this study).
Furthermore, it would be interesting and informative to know the extent of pathological lymph node metastasis (isolated tumor cells, micrometastases, macrometastases, extranodal extension). In this regard I invite the authors to include the recently published results of the SENOMAC trial (de Boniface NEJM 2024) in their discussion, as this trial now extended the use of SLNB also in patients with up to extranodal disease.
Lastly, the authors do not mention the TAXIS trial (NCT03513614), which includes patients with cN0/iN+ disease in the upfront surgical setting. The surgical concept used is tailored axillary surgery (TAS), consisting of the removal of the sentinel lymph nodes, the biopsy-prven node, and all palpably suspicious findings. Patients are then randomized to undergo completion axillary dissection (controal arm) or axillary radiotherapy (e.g., https://pmc.ncbi.nlm.nih.gov/articles/PMC10695869/). Please include this in your discussion as a study that will provide level I evidence on the question, whether ALND is necessary in these patients.
Minor comments:
Reference 15 on page 7, line 168 seems not to be formatted correctly.
On page 7 lines 176 and 208 please also mention the recent INSEMA trial (NEJM 2025).
Reference 18 seems not to be formatted correctly in the reference list.
Author Response
Comment 1: I was positively surprised to see that apparently the number of suspicious lymph nodes on ultrasonography (<3 nodes vs. >=3 nodes) were recorded in the registry as they are not recorded in many larger cancer centers.
If the data are available to the authors, I would find it interesting to know, how many patients received a breast MRI and/or PET-CT; again, of as much detailed information concerning those imaging modalities would be available as it seems to be the case for axillary ultrasound, it would be interesting to know, whether results of these diagnostic tools would predict higher nodal burden (however, this could be a novel study and therefore be out of scope for this study).
Answer 1: The registry included nodal assessments from the initial pretreatment evaluation, documenting the presence or absence of suspicious or involved lymph nodes in specific anatomical regions (infraclavicular, supraclavicular, internal mammary, and axillary), as well as the clinical status of axillary nodes (negative, positive, or matted). Additional variables captured whether patients had a suspicious node on ultrasound and a confirmed positive node on core needle biopsy. This allowed us to identify patients and patient disease characteristics. However specific axillary US characteristics were collected from retrospective chart review on these identified patients. Findings on MRI and PET scan were not collected and therefore unfortunately not available. We have added this as a limitation.
Comment 2: It would be interesting and informative to know the extent of pathological lymph node metastasis (isolated tumor cells, micrometastases, macrometastases, extra nodal extension). In this regard I invite the authors to include the recently published results of the SENOMAC trial (de Boniface NEJM 2024) in their discussion, as this trial now extended the use of SLNB also in patients with up to extra nodal disease.
Answer 2: The authors unfortunately did not collect this pathological data only number of nodes positive. We have mentioned the SENOMAC trial in our discussion and this limitation.
Comment 3: The authors do not mention the TAXIS trial (NCT03513614), which includes patients with cN0/iN+ disease in the upfront surgical setting. The surgical concept used is tailored axillary surgery (TAS), consisting of the removal of the sentinel lymph nodes, the biopsy-proven node, and all palpably suspicious findings. Patients are then randomized to undergo completion axillary dissection (control arm) or axillary radiotherapy (e.g., https://pmc.ncbi.nlm.nih.gov/articles/PMC10695869/). Please include this in your discussion as a study that will provide level I evidence on the question, whether ALND is necessary in these patients.
Answer 3: Paragraph added in discussion
Comment 4: Reference 15 on page 7, line 168 seems not to be formatted correctly.
Answer 4: Corrected
Comment 5: On page 7 lines 176 and 208 please also mention the recent INSEMA trial (NEJM 2025).
Answer: corrected
Comment 6: Reference 18 seems not to be formatted correctly in the reference list.
Answer: Corrected

Reviewer 2 Report
Comments and Suggestions for Authors
The article is a well-reported evaluation of early breast cancers (cN0 T1-3) within the Alberta database. I have one comment: could the authors report on the role of radiotherapy in this analyzed population? Specifically, did patients with 1 or 2 positive nodes receive axillary radiotherapy? I also suggest including a discussion on the potential role of newer therapies (e.g., ribociclib) in cases with sentinel node biopsy (SNB) and 1 or 2 positive nodes.
Author Response
The article is a well-reported evaluation of early breast cancers (cN0 T1-3) within the Alberta database. I have one comment: could the authors report on the role of radiotherapy in this analyzed population? Specifically, did patients with 1 or 2 positive nodes receive axillary radiotherapy? I also suggest including a discussion on the potential role of newer therapies (e.g., ribociclib) in cases with sentinel node biopsy (SNB) and 1 or 2 positive nodes.
We appreciate the reviewer’s interest in the role of radiotherapy in our cohort. However, radiation therapy data was not included in our data collection. Additionally, there was no institutional or standardized protocol for axillary radiation in this patient population during the study period. Decisions regarding regional nodal irradiation would have been made at the discretion of the individual radiation oncologist, based on evolving evidence and clinical judgment at the time of treatment. As such, the use of axillary radiation in patients with 1 or 2 positive nodes would have been heterogeneous and difficult to analyze meaningfully within the scope of this study.
Discussion on NATALEE and its implications added.
Reviewer 3 Report
Comments and Suggestions for Authors
I would like to thank the authors for addressing a highly relevant and timely topic in breast surgical oncology. The question of how to best manage axillary staging in the post-Z0011 era, particularly for patients with positive findings on preoperative axillary imaging, remains clinically significant.
However, I have a few methodological concerns that I would encourage the authors to address in a revised version of the manuscript.
-
Statistical Modeling
The use of logistic regression to predict high axillary burden is appropriate; however, I wonder why a propensity score matching approach was not considered to better control for potential confounders and further validate the model's robustness. The rationale for relying solely on logistic regression should be clarified. -
Novelty and Contribution to the Literature
While the study supports existing findings, the manuscript would benefit from a clearer articulation of what new insights it contributes to the current literature. What differentiates this study from previous research (e.g., Morrow, Boland, Caudle, et al.)? -
Figures and Tables
Acronyms and variables should be consistently defined within each figure and table legend. For example, BMI categories should include the numeric thresholds used to define “normal weight,” “overweight,” “obese,” etc., to ensure clarity and reproducibility. -
References
The references are generally valid, but several are outdated. The authors are encouraged to include more recent literature, particularly from the last 2–3 years, which could provide additional context or support for their conclusions. A few references from 2023 are mentioned, but expanding this set would strengthen the paper's positioning.
Lastly, the authors are encouraged to better elaborate on the practical clinical implications of their findings. What might realistically change in daily surgical practice based on their results?
In particular, if the goal is to propose a risk stratification model to guide axillary lymph node dissection (ALND), this model should rely exclusively on preoperative data, as it would be used to support decision-making before surgery. Clarifying this point would enhance the clinical relevance and applicability of the study.
In conclusion, the manuscript explores an important clinical issue and may be of interest to the journal’s audience—particularly those involved in surgical education and clinical decision-making in breast cancer management. I commend the authors for their efforts, and I encourage them to further clarify their methodology and highlight the original contributions of their work.
Author Response
Comment 1: use of logistic regression to predict high axillary burden is appropriate; however, I wonder why a propensity score matching approach was not considered to better control for potential confounders and further validate the model's robustness. The rationale for relying solely on logistic regression should be clarified.
Answer 1: While we agree that propensity score methods can be useful in controlling for confounding, our study was limited by a relatively small sample size (n=107), which would have significantly reduced the statistical power had we applied matching techniques. Propensity score matching can lead to case exclusion and loss of information when suitable matches are not available, particularly in small cohorts. Therefore, we opted to use multivariable logistic regression with forward stepwise selection to adjust for key confounding variables while retaining the full analytic sample.
Comment 2: While the study supports existing findings, the manuscript would benefit from a clearer articulation of what new insights it contributes to the current literature. What differentiates this study from previous research (e.g., Morrow, Boland, Caudle, et al.)?
Answer 2: paragraphs added to discussion
Comment 3: Acronyms and variables should be consistently defined within each figure and table legend. For example, BMI categories should include the numeric thresholds used to define “normal weight,” “overweight,” “obese,” etc., to ensure clarity and reproducibility
Answer 2: Corrected
Comment 4: The references are generally valid, but several are outdated. The authors are encouraged to include more recent literature, particularly from the last 2–3 years, which could provide additional context or support for their conclusions. A few references from 2023 are mentioned, but expanding this set would strengthen the paper's positioning.
Answer 4: Additional references and citations added
Comment 5: Lastly, the authors are encouraged to better elaborate on the practical clinical implications of their findings. What might realistically change in daily surgical practice based on their results? If the goal is to propose a risk stratification model to guide axillary lymph node dissection (ALND), this model should rely exclusively on preoperative data, as it would be used to support decision-making before surgery. Clarifying this point would enhance the clinical relevance and applicability of the study.
Answer 5: We agree that for a risk stratification model to have meaningful clinical utility, it must rely exclusively on preoperative data to guide surgical decision-making. Our model was intentionally built using only variables available prior to surgery—including patient demographics, tumor characteristics, and detailed preoperative ultrasound findings—all of which are routinely collected in standard clinical practice. This approach aimed to provide a practical, non-invasive method to predict high axillary burden. However, our results demonstrated that preoperative variables alone were not sufficiently predictive of high nodal burden. Therefore, we conclude that surgical management should continue to follow current NCCN and ASCO guidelines, with decisions regarding axillary surgery based on established criteria rather than risk models derived from preoperative imaging and clinical factors alone. This was more clearly defined in our discussion
Round 2
Reviewer 3 Report
Comments and Suggestions for Authors
Dear Authors,
Thank you for your thoughtful revisions and for clearly addressing the previous comments. I appreciate your efforts in improving the manuscript and clarifying several key points raised during the first round of review.
I would like to offer two additional suggestions that may further enhance the clarity and transparency of your study:
Flow diagram
To improve clarity and reader comprehension, I recommend adding a figure that illustrates the patient selection process and final classification based on axillary nodal burden. A flow diagram showing the initial cohort (n = 2947), the exclusion criteria, the final analytic sample (n = 107), and the distribution into groups with 1–2 vs ≥3 positive lymph nodes (as well as the T1–T2 subgroup and potential applicability of Z0011 criteria) would greatly strengthen the presentation and interpretability of your findings.
Table and figure legends
To enhance reproducibility and accessibility—particularly for international readers—I strongly recommend ensuring that all abbreviations and categorical variables used in tables and figures are clearly defined in the captions or included as footnotes. For example, abbreviations such as BCS (breast-conserving surgery), LVI (lymphovascular invasion), BMI (body mass index), and HER2 (human epidermal growth factor receptor 2) should be spelled out at first mention or explicitly listed in a legend.
Moreover, categorical groupings (e.g., BMI ranges, tumor grade, and ultrasound findings such as “replaced hilum” or “thickened cortex”) should include short definitions or classification thresholds. Numeric labels for “facility” (1–13) would also benefit from clarification—specifying whether these refer to hospitals, radiology centers, or individual operators.
Including this additional information will significantly improve the clarity, readability, and scientific transparency of your results.
Thank you again for your valuable contribution to the field.
Author Response
Flow diagram
To improve clarity and reader comprehension, I recommend adding a figure that illustrates the patient selection process and final classification based on axillary nodal burden. A flow diagram showing the initial cohort (n = 2947), the exclusion criteria, the final analytic sample (n = 107), and the distribution into groups with 1–2 vs ≥3 positive lymph nodes (as well as the T1–T2 subgroup and potential applicability of Z0011 criteria) would greatly strengthen the presentation and interpretability of your findings. --> added as Figure 2
Table and figure legends
To enhance reproducibility and accessibility—particularly for international readers—I strongly recommend ensuring that all abbreviations and categorical variables used in tables and figures are clearly defined in the captions or included as footnotes. For example, abbreviations such as BCS (breast-conserving surgery), LVI (lymphovascular invasion), BMI (body mass index), and HER2 (human epidermal growth factor receptor 2) should be spelled out at first mention or explicitly listed in a legend. --> Table legend add as footnote
Moreover, categorical groupings (e.g., BMI ranges, tumor grade, and ultrasound findings such as “replaced hilum” or “thickened cortex”) should include short definitions or classification thresholds. Numeric labels for “facility” (1–13) would also benefit from clarification—specifying whether these refer to hospitals, radiology centers, or individual operators. --> added where appropriate